# DHX15 regulates CMTR1-dependent gene expression and cell proliferation

Francisco Inesta-Vaquera[1], Viduth K Chaugule[2], Alison Galloway[1], Laurel Chandler[3], Alejandro Rojas-Fernandez[4], Simone Weidlich[5], Mark Peggie[5], Victoria H Cowling[1]

CMTR1 contributes to mRNA cap formation by methylating the first transcribed nucleotide ribose at the O-2 position. mRNA cap O-2 methylation has roles in mRNA stabilisation and translation, and self-RNA tolerance in innate immunity. We report that CMTR1 is recruited to serine-5–phosphorylated RNA Pol II C-terminal domain, early in transcription. We isolated CMTR1 in a complex with DHX15, an RNA helicase functioning in splicing and ribosome biogenesis, and characterised it as a regulator of CMTR1. When DHX15 is bound, CMTR1 activity is repressed and the methyltransferase does not bind to RNA pol II. Conversely, CMTR1 activates DHX15 helicase activity, which is likely to impact several nuclear functions. In HCC1806 breast carcinoma cell line, the DHX15–CMTR1 interaction controls ribosome loading of a subset of mRNAs and regulates cell proliferation. The impact of the CMTR1–DHX15 interaction is complex and will depend on the relative expression of these enzymes and their interactors, and the cellular dependency on different RNA processing pathways.

## Introduction

Formation of the mRNA cap initiates the maturation of RNA pol II transcripts into translation-competent mRNA (Furuichi, 2015). The mRNA cap protects transcripts from degradation and recruits protein complexes involved in nuclear export, splicing, 3′ processing, and translation initiation (Topisirovic et al, 2011; Ramanathan et al, 2016). mRNA cap formation initiates with the addition of an inverted guanosine group, via a tri-phosphate bridge, to the first transcribed nucleotide of nascent RNA pol II transcripts. Subsequently, this guanosine cap is methylated on the N-7 position to create the cap 0 structure, which binds efficiently to CBC, eIF4F, and other complexes involved in RNA processing and translation initiation. The initial transcribed nucleotides are further methylated at several other positions in a species-specific manner. In mammals, the O-2 position of the riboses of the first and second transcribed nucleotides are sites of abundant methylation (Langberg & Moss, 1981).

A series of enzymes catalyse mRNA cap formation, which have different configurations in different species (Shuman, 2002). In mammals, RNGTT/capping enzyme catalyses guanosine cap addition and RNA guanine-7 methyltransferase (RNMT)-RNMT-activating miniprotein (RAM) catalyses guanosine cap N-7 methylation. RNGTT/capping enzyme and RNMT-RAM are recruited to RNA pol II at the initiation of transcription (Buratowski, 2009). CMTR1 and CMTR2 methylate the O-2 position of first and second transcribed nucleotide riboses, respectively (Belanger et al, 2010; Werner et al, 2011; Inesta-Vaquera & Cowling, 2017). *CMTR1* (ISG95, FTSJD2, KIAA0082) was first identified as a human-interferon–regulated gene (Su et al, 2002; Geiss et al, 2003; Guerra et al, 2003; Kato et al, 2003). It was recognised to have several functional domains including a methyltransferase domain (Haline-Vaz et al, 2008). Subsequently, CMTR1 was biochemically characterised as the O-2 ribose methyltransferase of the first transcribed nucleotide and the catalytic domain was crystalized with *S*-adenosyl methionine (SAM) and a capped oligonucleotide (Belanger et al, 2010; Smietanski et al, 2014).

CMTR1 and first nucleotide O-2 methylation have functions in gene expression and innate immunity. O-2 methylation is likely to influence recruitment of cap-binding protein complexes and promote ribosomal subunit binding (Muthukrishnan et al, 1976b). Following fertilization of sea urchin eggs, N-7 and O-2 cap methylation is associated with translational up-regulation of a subset of maternal transcripts (Caldwell & Emerson, 1985). During *Xenopus laevis* oocyte maturation, first nucleotide O-2 methylation significantly increases translation efficiency and is required for the translation of maternal mRNA (Kuge & Richter, 1995; Kuge et al, 1998). Recently, cap O-2 methylation was demonstrated to be critical for preventing decapping exoribonuclease-mediated decapping, which leads to RNA degradation (Picard-Jean et al, 2018). In mice, a significant proportion of the first nucleotides were found to be O-2 methylated on the ribose, although the relative proportion of this methylation varied between organs, indicating a regulated event (Kruse et al, 2011).

[1]Centre for Gene Regulation and Expression, School of Life Sciences, University of Dundee, Dundee, UK    [2]Institute of Molecular, Cell and Systems Biology, School of Life Sciences, University of Glasgow, Glasgow, UK    [3]Nuffield Department of Clinical Neurosciences, Medical Sciences Division, University of Oxford, Oxford, UK    [4]Center for Interdisciplinary Studies on the Nervous System and Institute of Medicine, Universidad Austral de Chile, Valdivia, Chile    [5]Division of Signal Transduction Therapies, School of Life Sciences, University of Dundee, Dundee, UK

Correspondence: v.h.cowling@dundee.ac.uk

The composition of the 5′ cap is also an important determinant of self- (host) versus non–self-RNA during viral infection (Leung & Amarasinghe, 2016). The absence of O-2 methylation in viral transcripts results in enhanced sensitivity to the interferon-induced IFIT proteins; first nucleotide O-2 methylation distinguishes self from non–self-RNA (Daffis et al, 2010). CMTR1-dependent O-2 methylation abrogates the activation of retinoic acid inducible gene I, a helicase that initiates immune responses on interaction with uncapped or aberrantly capped transcripts (Schuberth-Wagner et al, 2015).

Here, we report the first regulator of CMTR1 function. We demonstrate that CMTR1 and the DEAH (Asp-Glu-Ala-His)-box RNA helicase, DHX15, form a stable complex in cells and reciprocally influence activity and action. DHX15 reduces CMTR1 methyltransferase activity. CMTR1 activates DHX15 helicase activity and influences nuclear localisation. Disruption of the CMTR1–DHX15 interaction leads to increased ribosome loading of a subset of mRNAs involved in key metabolic functions and impacts on cell proliferation.

# Results

## CMTR1 interacts directly with DHX15

To investigate the regulation and function of CMTR1, we identified CMTR1-interacting proteins. HA-CMTR1 was immunoprecipitated from HeLa cell extracts and resolved by SDS–PAGE, and co-purified proteins were identified by mass spectrometry (Fig 1A). DHX15 (O43143), a 95-kD DEAH-box RNA helicase, was the only protein identified with significant mascot scores and coverage in HA-CMTR1 immunoprecipitates (IP) (Fig S1) (Imamura et al, 1997). Conversely, CMTR1 was identified in HA-DHX15 IPs using mass spectrometry (Figs 1B and S1). To verify their interaction, GFP-CMTR1 and FLAG-DHX15 were expressed in HeLa cells and co-immunoprecipitated from cell extracts (Fig 1C). To investigate endogenous CMTR1, an antibody was raised against recombinant human CMTR1, which detected CMTR1 in HeLa cell extracts using Western blot and immunoprecipitation (IP) (Fig S2A and B). The interaction of endogenous CMTR1 and DHX15 was confirmed using IP in HeLa, HCC1809, U2OS, and HEK293 cell extracts (Fig 1D). In *CMTR1* null HeLa cells, the anti-CMTR1 antibody was unable to purify DHX15, discounting non-specific interactions of DHX15 with resin or antibody (Fig 1E). Demonstrating the specificity of the CMTR1–DHX15 interaction, DHX16, another DEAH RNA helicase, was not detected in CMTR1 IPs, and RNMT, another cap methyltransferase, was not detected in DHX15 IPs (Fig S2C and D). Furthermore, the CMTR1–DHX15 interaction was sustained on RNaseA treatment and therefore is not maintained by an RNA connector (Fig 1F). In HeLa cells, DHX15 and CMTR1 are expressed at equivalent molar ratios (Nagaraj et al, 2011). Equimolar recombinant human CMTR1 and His$^6$-DHX15 co-immunoprecipitated, confirming their direct interaction (Figs 1G and S2E). Neither CMTR1 or DHX15 homodimerise (Fig S2F and G).

To gain insight into CMTR1 and DHX15 complexes in HeLa cells, gel filtration analysis was performed. Used as markers, recombinant CMTR1 and His$^6$-DHX15 monomers migrated as expected for ~100 kD monomers (Fig 1H, lower panels, 14.5 ml). HeLa extracts were treated with RNases I and A before gel filtration to prevent protein–protein interaction via a RNA linker (Figs 1H, upper panels, and S2H). Cellular

CMTR1 and DHX15 co-eluted after the 158 kD marker consistent with DHX15–CMTR1 heterodimers. As described above, no other interacting proteins were identified in mass spectrometry analysis of CMTR1 IPs (Figs 1A and S1). Conversely, DHX15, was also observed in larger, CMTR1-independent complexes by gel filtration (Fig 1H, 11.5 ml). Mass spectrometry analysis of DHX15 IPs identified a series of other proteins, including some with a G-patch domain. Novel and all previously reported DHX15 interactors were identified, including those involved in ribosomal biogenesis and splicing (Figs 1B and S3). This result reflects the well-documented function of DHX15 and its homologues in these processes (Robert-Paganin et al, 2015; Memet et al, 2017). Although these other DHX15 complexes are biologically interesting, for the remainder of this study we focus on the relationship between DHX15 and CMTR1.

## CMTR1 G-patch domain interacts with the DHX15 oligonucleotide/oligosaccharide binding (OB)–fold

CMTR1 is an 835 residue protein containing an NLS (residues 2–19), a G-patch domain (residues 85–133), an RrmJ-type SAM-dependent O-2 methyltransferase domain (RFM, residues 170–550), a guanylyltransferase-like domain (GT-like, residues 560–729), and a WW domain (755–790) (Fig 2A) (Aravind & Koonin, 1999; Haline-Vaz et al, 2008; Belanger et al, 2010; Smietanski et al, 2014). To map the interacting regions of CMTR1 and DHX15, HeLa cells were transfected with GFP-CMTR1 wild-type (WT) or mutants ΔN (25–835), ΔG (143–835), 1–143, and Gp (85–143), or GFP alone (Fig 2A). Endogenous DHX15 co-immunoprecipitated with GFP-CMTR1 WT and mutants except GFP-ΔG, the G-patch deletion mutant (Fig 2B). Furthermore, DHX15 co-immunoprecipitated with GFP-Gp, the G-patch domain of CMTR1, indicating that this domain mediates the interaction.

DHX15 is a prototypic member of the DEAH family of RNA helicases (Jankowsky, 2011). It contains an N-terminal domain of unknown function (N-term, residues 1–146), two Rec-A tandem repeats (Rec A1–A2, residues 147–518), a WH domain (residues 519–572), a ratchet domain (residues 572–671), and an OB-fold (residues 671–795) (Fig 2C). The OB-fold is the predominant site of interaction with RNA and proteins, although the Rec-A domains can also establish functional interactions with RNA and protein (Lebaron et al, 2009). Other G-patch–containing proteins have been demonstrated to interact with DHX15 via the OB-fold (Fig S3) (Lin et al, 2009; Niu et al, 2012; Chen et al, 2014; Memet et al, 2017). To determine whether the DHX15 OB-fold is required for CMTR1 binding, HA-DHX15 WT or C-terminal deletion mutant (ΔC, 1–635) were expressed in HeLa cells. Endogenous CMTR1 immunoprecipitated with HA-DHX15 WT but not ΔC (Fig 2D).

G-patch domains have the consensus sequence, hhxxxGaxxGxGhGxxxxG; "G" is glycine, "h" is a bulky, hydrophobic residue (I, L, V, M), "a" is an aromatic residue (F, Y, W), and "x" is any residue (Aravind & Koonin, 1999). The CMTR1 G-patch has this consensus and in addition leucines at residues 94, 106, and 128 which are conserved in the G-patch domains of the established DHX15 interactors, PINX1, NKRF, GPATCH2, and RBM5 (Fig 2E) (Lin et al, 2009; Niu et al, 2012; Chen et al, 2014; Memet et al, 2017). Because these conserved leucines in other DHX15-binding proteins are required for the interaction, the impact of mutating CMTR1 L94, L106, and L128 to alanine was investigated using the 2L/A mutant (L94A and L106A),

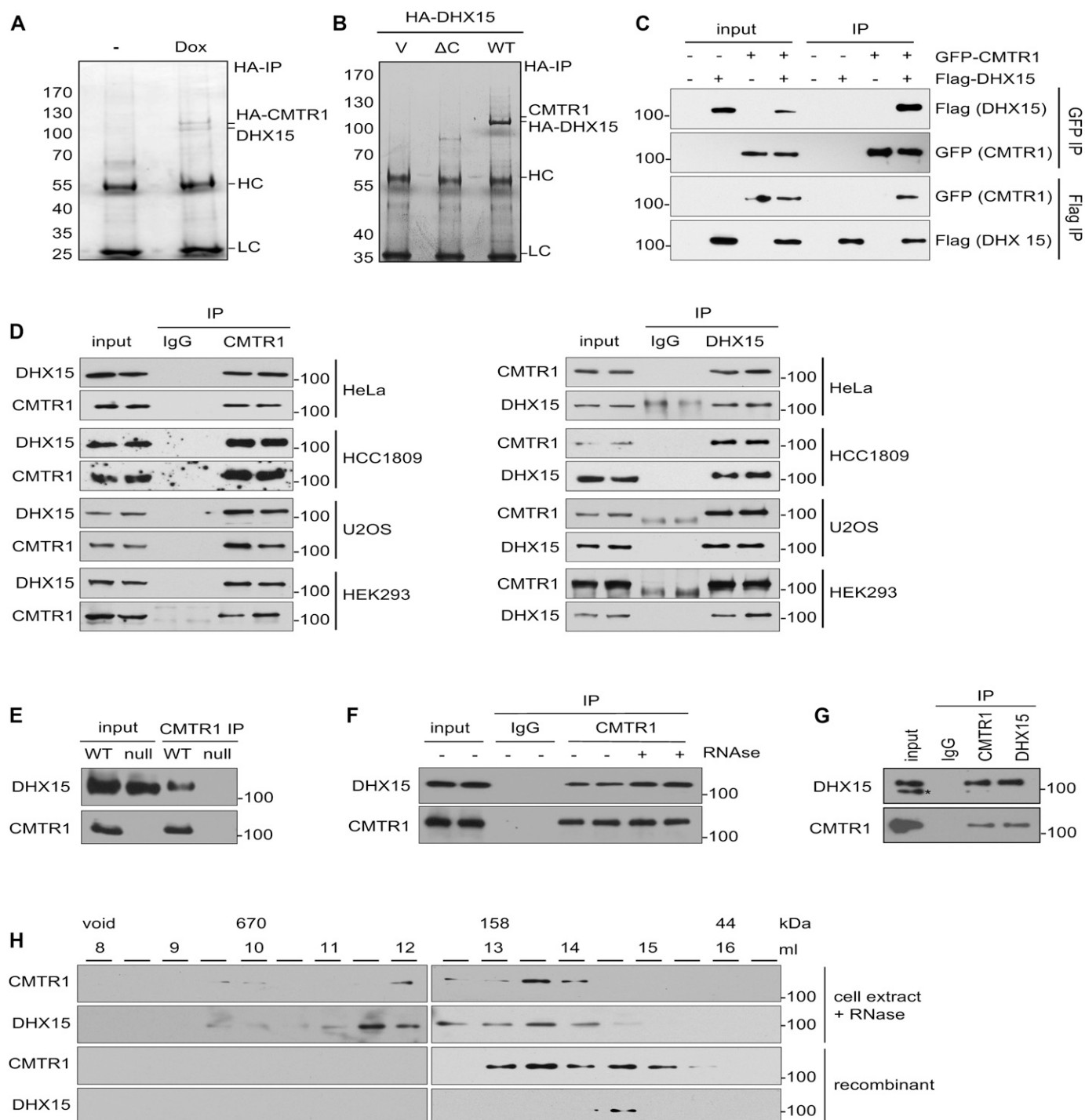

**Figure 1.   DHX15 interacts with CMTR1.**
**(A)** HA-CMTR1 was doxycyline-induced (Dox) in a HeLa cell line. Anti-HA antibody immunoprecipitation (IP) was performed on cell extracts. IPs were resolved by SDS–PAGE and Coomassie Blue-stained. HA-CMTR1, DHX15, antibody heavy chain (HC), and light chain (LC) are indicated. **(B)** pcDNA5 HA-DHX15 WT, 1-635 (ΔC), and vector control (V) were transfected into HeLa cells. Anti-HA IP was performed and analysed as in (A). **(C)** pcDNA5 GFP-CMTR1 and FLAG-DHX15 were expressed in HeLa cells. IPs were performed on cell extracts using anti-GFP and FLAG antibodies. **(D)** Endogenous CMTR1 (left panels) or DHX15 (right panels) was immunoprecipitated from extracts of the cell lines indicated; Western blot analysis. Sheep IgG was used for IP control. **(E)** In HeLa cells, transfection of guide RNAs and Cas9 resulted in a *CMTR1* null cell line. Endogenous CMTR1 IP was performed on extracts of HeLa cells (WT) or *CMTR1* null HeLa cells (null). **(F)** CMTR1 IPs from HeLa cells were untreated or incubated with RNAse A prior to analysis. **(G)** 100 ng recombinant CMTR1 and 100 ng His6-DHX15 were mixed and subjected to IP with indicated antibodies. *denotes non-specific band. **(H)** HeLa cell extracts were treated with RNaseI and RNaseA. Extracts, recombinant CMTR1, or recombinant His6-DHX15 were resolved on a Superdex s200 10/30 column. Fractions were analysed by Western blot. Elution of standards indicated.

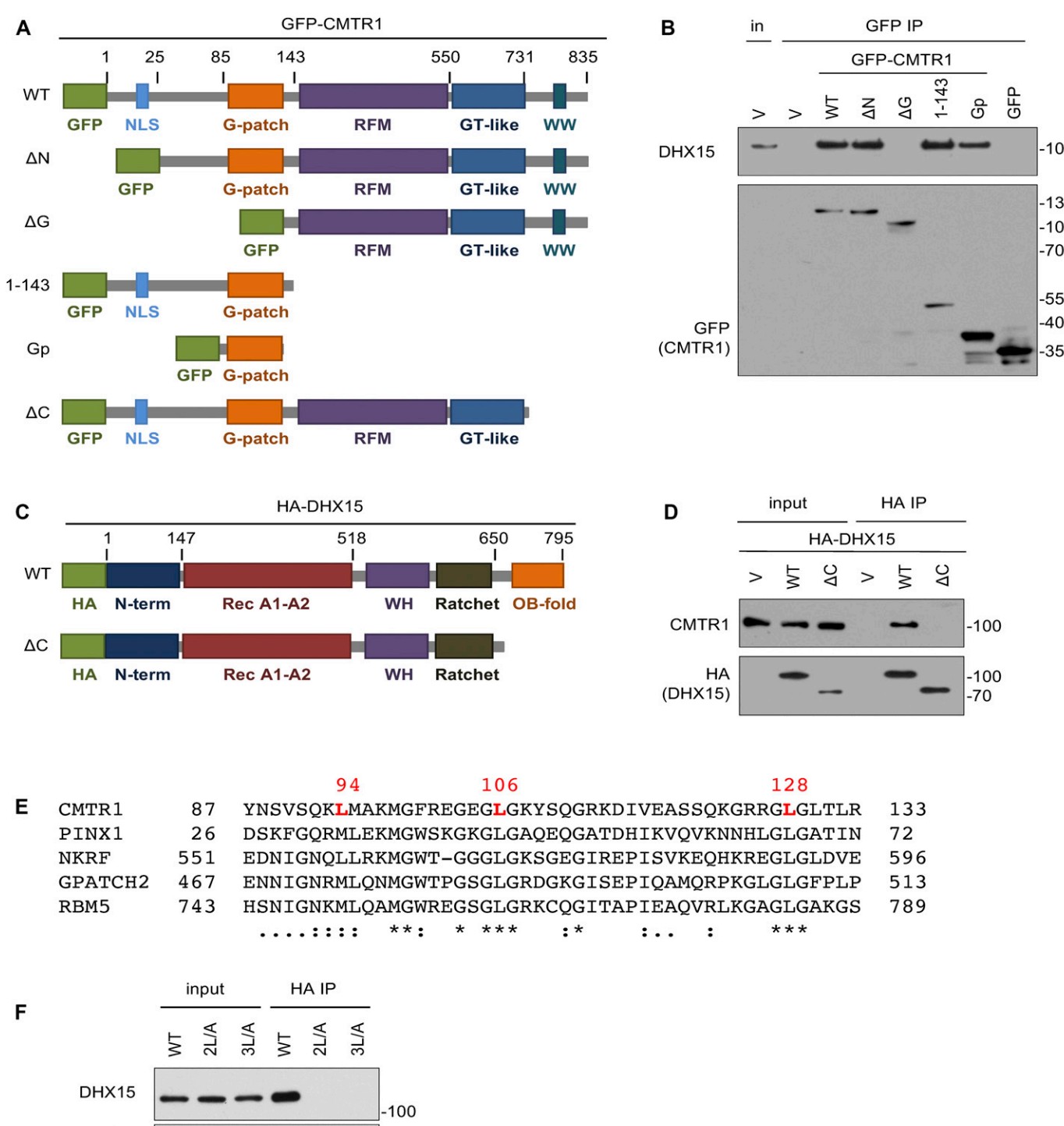

**Figure 2. CMTR1 G-patch domain binds to DHX15 OB-fold domain.**
**(A)** Diagram of CMTR1 domains and mutants made. **(B)** HeLa cells were transfected with pcDNA5 (V), or with pcDNA5 GFP, GFP-CMTR1 WT, and mutants. Anti-GFP-antibody IPs (GFP-IP) were performed and analysed by Western blot. Inputs were also analysed (in). **(C)** Diagram of DHX15 domains and mutants made. **(D)** HeLa cells were transfected with pcDNA5 (V), pcDNA5 HA-DHX15, and mutants. Anti-HA antibody IPs (HA-IP) were performed and analysed by Western blot. **(E)** Alignment of G-patch domains in DHX15 interactors generated by Clustal Omega Alignment tool. Conserved leucines in CMTR1 in red. Residues: ".," weakly similar; ":," strongly similar; "*," conserved. **(F)** HeLa cells were transfected with pcDNA5 HA-CMTR1 WT and mutants. HA-IPs were performed and analysed by Western blot.

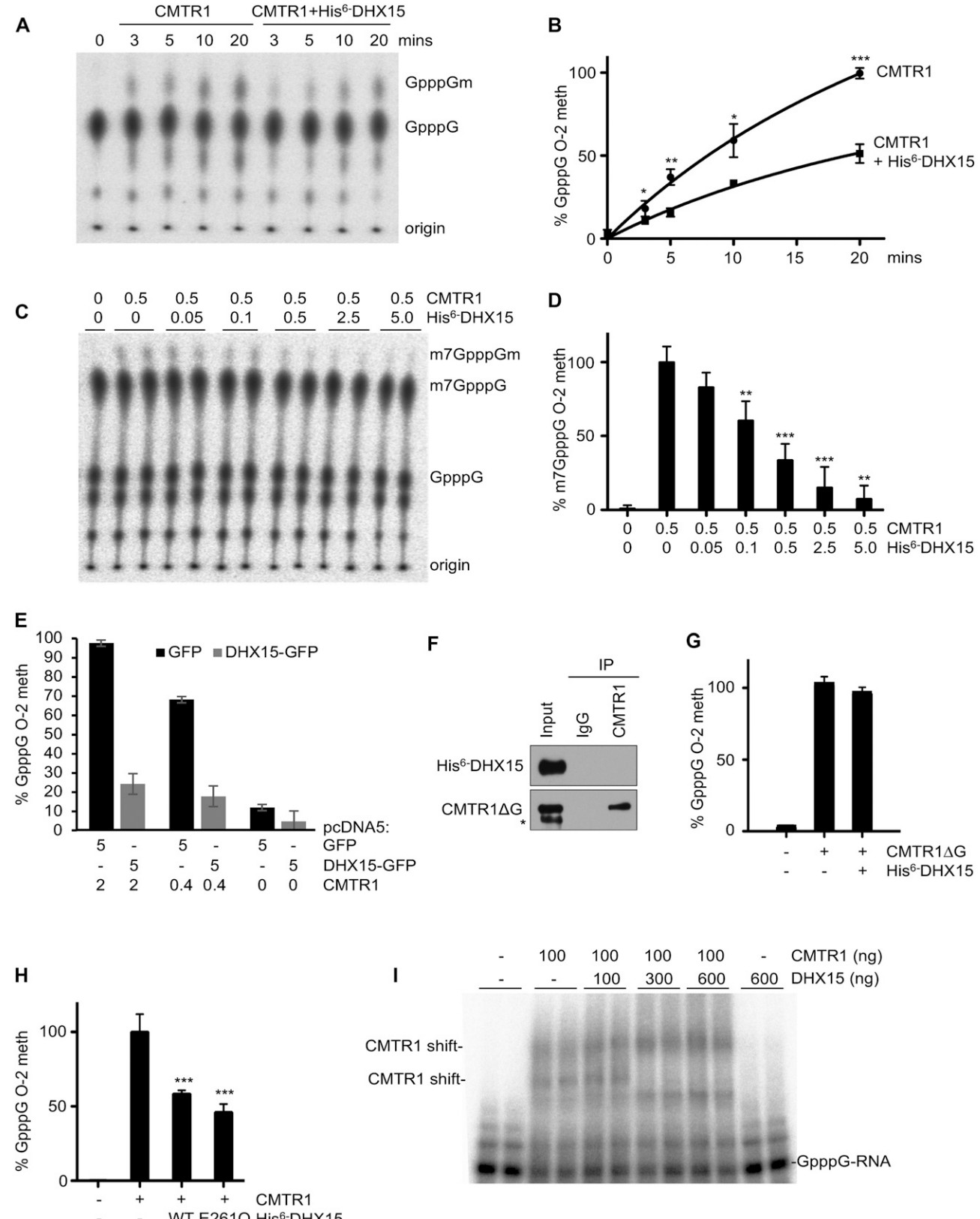

**Figure 3. CMTR1 methyltransferase activity is repressed by DHX15.**
**(A)** GpppG-capped RNA was incubated with 3 nM CMTR1 or 3 nM CMTR1 and 3 nM His[6]-DHX15 for the time indicated. Caps throughout the figure are [32]P-labelled on α-phosphate. Generated GpppGm (first transcribed nucleotide ribose 0–2 methylated) resolved by thin-layer chromatography and detected by autoradiography. **(B)**

and the 3L/A mutant (L94A, L106A, and L128A) (Fig 2E). The 2L/A mutation was sufficient to abrogate the interaction of CMTR1 with DHX15, confirming that CMTR1 interacts with DHX15 through the G-patch domain (Fig 2F).

### DHX15 inhibits CMTR1 methyltransferase activity

To investigate the biochemical impact of the CMTR1–DHX15 interaction, we first analysed CMTR1 methyltransferase activity. A [32]P-labelled, guanosine-capped transcript (GpppG) was incubated with recombinant CMTR1 and a methyl donor, SAM (Fig 3A). First transcribed nucleotide O-2 methylation (GpppG to GpppGm conversion) was observed by resolution on thin-layer chromatography (Belanger et al, 2010). Incubation of recombinant CMTR1 with equimolar recombinant His[6]-DHX15 resulted in an approximately 50% reduction in O-2 methylation (Fig 3A and B). Similarly, His[6]-DHX15 significantly inhibited O-2 methylation of a 7-methylguanosine-capped transcript (m7GpppG to m7GpppGm conversion) (Figs 3C and D, and S4A and B). His[6]-DHX15 inhibited CMTR1-dependent methylation in a dose-dependent manner (Fig 3C and D). The impact of DHX15 on CMTR1-methyltransferase activity was also observed in cells. Transfection of HeLa cells with a titration of pCDNA5 HA-CMTR1 resulted in dose-dependent O-2 methyltransferase activity in cell extracts (Fig 3E). Transfection with pCDNA5 GFP-DHX15 inhibited methyltransferase activity in cell extracts. As controls, ATP did not alter DHX15-dependent repression of CMTR1, and BSA and recombinant protein storage buffer did not affect CMTR1-dependent methylation (Fig S4C–E).

To investigate whether the interaction of DHX15 and CMTR1 is required for inhibition of methyltransferase activity, CMTR1ΔG mutant, which does not interact with DHX15, was utilised (Figs 2A and B, and 3F). Recombinant CMTR1ΔG had slightly reduced activity compared with WT (Fig S4F); however, its methyltransferase activity was not inhibited by DHX15 (Fig 3G). This indicates that direct binding of DHX15 is required for the inhibition of CMTR1 methyltransferase activity. Because DHX15 is a helicase, we determined the requirement for its catalytic activity to inhibit CMTR1 methyltransferase activity. Incubation of recombinant CMTR1 with equimolar recombinant His[6]-DHX15 WT or E261Q (Memet et al, 2017), a catalytically inactive mutant, resulted in an equivalent reduction in methyltransferase activity (Fig 3H). Therefore, inhibition of CMTR1 activity is a non-catalytic function of DHX15.

To gain insight into the mechanism of CMTR1 inhibition by DHX15, we investigated the impact on RNA binding. Unfortunately, we were unable to purify CMTR1 bound to RNA in cells using cross-linking methodologies which however detected RNMT–RNA interactions (Varshney et al, 2018). This may reflect the dynamic nature of CMTR1–RNA interactions in cells. However, recombinant CMTR1 could be detected interacting with GpppG-RNA, visualised as mobility shifts on a non-denaturing acrylamide gel (Fig 3I). Two major mobility CMTR1-RNA shifts were observed. Because the recombinant CMTR1 used in these studies was highly purified (Fig S2E), two mobility shifts is likely to indicate conformational isomers in CMTR1-RNA complexes. An interaction between His[6]-DHX15 alone and GpppG-RNA was not observed. In the presence of increasing amounts of DHX15, the CMTR1 GpppG-RNA complex was not reduced and further bands were observed, indicating GpppG-RNA-CMTR1-DHX15 complexes. Although in vitro, these RNA-protein interaction studies indicate that DHX15 does not reduce the affinity of CMTR1 for the GpppG-RNA substrate.

### CMTR1 increases DHX15 helicase activity

To determine the impact of the CMTR1–DHX15 interaction, we also investigated whether CMTR1 influences DHX15 ATPase and helicase activity (Fig 4). Recombinant His[6]-DHX15 was incubated with $\alpha$[32]P-ATP and hydrolysis visualised by the detection of $\alpha$[32]P-ADP (Fig 4A and B). As established, addition of RNA significantly increased ATP hydrolysis (Fig 4A and B, and S5) (Walbott et al, 2010; Memet et al, 2017). Previously characterised G-patch–containing interactors of Prp43 or DHX15 have variable impact on ATPase activity (Tanaka et al, 2007; Lebaron et al, 2009; Niu et al, 2012). To determine whether CMTR1 influences DHX15-dependent ATP hydrolysis, a titration of recombinant CMTR1 was included in the ATPase assay. CMTR1 did not activate or repress basal DHX15-dependent ATP hydrolysis (Fig 4C and D). Furthermore, CMTR1 did not activate or repress RNA-stimulated DHX15-dependent ATP hydrolysis (Fig 4E). Addition of SAM had no impact on DHX15-dependent ATP hydrolysis, including in the presence of CMTR1 or RNA (Fig 4F).

Because interactors of Prp43, the yeast DHX15 homologue, can regulate helicase activity independently of ATPase activity, we investigated whether CMTR1 could regulate DHX15 helicase activity (Tanaka et al, 2007). A [32]P-DNA-RNA duplex was incubated with 1,000 nM His[6]-DHX15 and helicase activity was confirmed by the loss of the duplex (Fig 4G, compare lanes 2 and 7). When 100 nM His[6]-DHX15 was used in this assay, reduced helicase activity was observed, as expected (lane 3). Addition of CMTR1 increased helicase activity in a dose-dependent manner (Fig 4G, lanes 4–6, and H). CMTR1 alone had no helicase activity (Fig 4G, lane 8).

### CMTR1 interaction with DHX15 and RNA pol II CTD are mutually exclusive

CMTR1 interacts with RNA pol II, which may permit efficient co-transcriptional first nucleotide O-2 methylation (Haline-Vaz et al,

Percentage conversion of GpppG to GpppGm (% GpppG O-2 meth) is plotted. Average and standard deviation for three independent experiments are plotted here and throughout the figure. **(C)** m7GpppG-capped RNA was incubated with 0.5 nM CMTR1 and indicated nM His[6]-DHX15 for 30 min. Generated of m7GpppGm detected. **(D)** Average percentage of m7GpppG O-2 meth and standard deviation are plotted. **(E)** HeLa cells were transfected with 5 μg pcDNA GFP or GFP-DHX15 and 0, 0.4, or 2 μg pcDNA5 HA-CMTR1, as indicated. 1 d after transfection, cells were harvested and 5 μg cell extract was used in O-2 methyltransferase assay. Percentage of GpppG O-2 meth and standard deviation of duplicates are reported. **(F)** 100 ng recombinant GST-CMTR1ΔG and 100 ng His[6]-DHX15 were mixed and subjected to IP with indicated antibodies; Western blot analysis. *denotes non-specific band. **(G)** GpppG-capped RNA was incubated with 5 nM GST-CMTR1ΔG or 5 nM GST-CMTR1ΔG and 5 nM His[6]-DHX15. Average percentage of GpppG O-2 meth and standard deviation are plotted. **(H)** GpppG-capped RNA was incubated with 3 nM CMTR1 and 3 nM His[6]-DHX15 WT or E261Q. Average percentage of GpppG O-2 meth and standard deviation are plotted. Where relevant, t test was performed. *P-value < 0.05; **P-value < 0.01; ***P-value < 0.005. **(I)** 55 nt [32]P GpppG-RNA incubated with 100 ng CMTR1 and indicated ng His[6]-DHX15. Complexes were resolved in a 4%–20% tris-borate-EDTA buffer non-denaturing acrylamide gel and detected using a phosphorimager (GE healthcare Life Sciences). Migration of [32]P GpppG-RNA probe and complexes are indicated.

2008). We investigated the impact of DHX15 on CMTR1 recruitment to RNA pol II. The interaction of recombinant CMTR1 and DHX15 with a biotinylated peptide consisting of three RNA pol II C-terminal domain heptad repeats, (YSPTSPS)$_3$, unphosphorylated (CTD), or phosphorylated on serine-5 (pCTD), was investigated (Fig 5A–C). As a control, RNGTT was demonstrated to interact with pCTD but not CTD (Fig S6A) (Ho & Shuman, 1999). The CMTR1 monomer bound to pCTD but not to CTD (Fig 5A), whereas the His$^6$-DHX15 monomer did not bind to either peptide (Fig 5B). To investigate whether CMTR1 can recruit DHX15 to pCTD, the recombinant proteins were mixed before peptide pulldown. Although CMTR1 and DHX15 interact in vitro (Fig 1G), CMTR1 was recruited to pCTD, whereas DHX15 was not, revealing that DHX15–CMTR1 complexes are not recruited to pCTD (Fig 5C).

The interaction of CMTR1 and DHX15 with RNA pol II was investigated in cells. In HeLa cell extracts, endogenous CMTR1 co-immunoprecipitated with RNA pol II phospho-Ser5-CTD (pSer5-CTD), but not phospho-Ser2-CTD (pSer2-CTD) or unphosphorylated RNA pol II (Fig 5D). The interaction of CMTR1 and pSer5-CTD was RNase A–insensitive and, therefore, RNA-independent (Fig S6B). To map the interaction of CMTR1 with RNA pol II, GFP-CMTR1 WT; 1–143, ΔG, and ΔC (1–173); and HA-CMTR1, 2L/A, and ΔC were expressed in HeLa cells and immunoprecipitated via their tags, and pSer5-CTD binding was determined (Fig 5E and F). GFP-CMTR1 and HA-CMTR1 interacted with pSer5-CTD, whereas GFP-CMTR1ΔC and HA-CMTR1ΔC did not, indicating that the WW domain mediates the interaction with RNA pol II (Fig 5E and F). GFP-CMTR1ΔG and HA-CMTR1 2L/A (both defective for DHX15 binding) interacted with pSer5-CTD, confirming that in cells DHX15 is not required for CMTR1 recruitment to RNA pol II (Fig 5E and F).

Although DHX15 is not required for CMTR1 recruitment to pSer5-CTD (Fig 5A, C, E, and F), it was important to determine whether DHX15 binds to RNA pol II in cells, either independently of CMTR1 or in a complex with it. HA-DHX15 was expressed in HeLa cells and immunoprecipitated via the HA tag (Fig 5G). As expected, HA-DHX15 bound to CMTR1 but not to pSer5-CTD or unphosphorylated RNA pol II. This confirms that in cells, as in vitro, DHX15 does not appreciably interact with RNA pol II (Fig 5B, C, and G) and, furthermore, that the CMTR1-DHX15 complex does not interact with RNA pol II (Fig 5C and G).

## CMTR1 influences DHX15 localisation

Following biochemical analysis of the CMTR1-DHX15 interaction, we investigated whether DHX15 influences CMTR1 localisation. We confirmed that GFP-CMTR1 and endogenous CMTR1 are predominantly nuclear in HeLa cells (Fig 6A–C) (Haline-Vaz et al, 2008). A potential CMTR1 nuclear localisation signal was identified, 14KKQKK (Lange et al, 2007). Mutation of CMTR1 14KKQKK to 14EEQEE (4K/E) or removal of the first 25 residues (ΔN) resulted in partial cytoplasmic localisation, confirming that 14KKQKK contributes to nuclear localisation. To determine the impact of DHX15 on CMTR1 localisation, the localisation of GFP-CMTR1 3L/A (does not bind DHX15) was investigated. GFP-CMTR1 3L/A was predominantly nuclear, indicating that DHX15 does not influence CMTR1 localisation (Fig 6A). Furthermore, suppression of DHX15 expression did not alter CMTR1 nuclear localisation (Fig 6B).

Because we had observed that CMTR1 stimulates DHX15 helicase activity in vitro, we investigated whether it also influences DHX15 localisation in cells. Prp43, the yeast homologue of DHX15, occupies several cellular locations to execute different biological functions (Heininger et al, 2016). The different Prp43 locations are achieved by the competition of G-patch proteins, which recruit the helicase to different parts of the cell. In HeLa cells, DHX15 exhibited a diffuse nuclear localisation (Fig 6B and C). However, in approximately a third of cells, DHX15 also exhibited speckled nuclear foci, a common feature of splicing factors (Tannukit et al, 2009). DHX15 foci partially co-localized with the splicing factor SC35 (Pawellek et al, 2014) (Fig S7A). The number of cells exhibiting DHX15 nuclear foci increased when CMTR1 was knocked down (Figs 6C and S7B) or in the cmtr1 null HeLa cell line (Fig S7C and D). The potential impact of CMTR1 on DHX15 localisation is explored in the Discussion section.

## DHX15 represses CMTR1-dependent translation

We had observed that DHX15 inhibits CMTR1-dependent O-2 methylation (Fig 3), CMTR1 activates DHX15-dependent helicase activity (Fig 4), and DHX15–CMTR1 complexes are not present on RNA pol II (Fig 5). The net outcome of the DHX15 and CMTR1 relationships is therefore complex and will depend on many factors (see the Discussion section). Here, we focussed our investigation on the impact of DHX15 on CMTR1 function using the CMTR1 2L/A mutant, which does not bind to DHX15 (Fig 2F), but is recruited to RNA pol II (Fig 5F). Because first nucleotide ribose O-2 methylation is associated with enhanced translation, the impact of the DHX15–CMTR1 interaction on this process was investigated (Kuge et al, 1998).

Initially we attempted to express CMTR1 WT and 2L/A mutant in CMTR1 null HeLa cells (Fig 1E). CMTR1 WT and 2L/A were expressed equivalently in the first 3 d after initial transduction. However, after selection, cells expressing the WT protein survived, whereas cells expressing the 2L/A mutant did not, indicating some growth disadvantage of the mutant. Therefore, CMTR1 2L/A was expressed in the presence of endogenous CMTR1. HCC1806 cells, a mammary epithelial tumour cell line, were selected for these experiments because we had previously established that these cells are highly sensitive to changes in the CMTR1 expression level. Cells were transfected with pcDNA5 HA-CMTR1 WT, 2L/A, or vector control and polysome profiling analysis was performed in which free ribosomes and ribosomal subunits are separated from translating ribosomes in a sucrose gradient (Fig 7A). Expression of HA-CMTR1 WT and 2/LA were equivalent (Fig 7B). Although expression of CMTR1 WT and 2L/A had a mild impact on the polysome profiles, in multiple experiments, the ratios between polysomes and monosomes were not significantly different, indicating that the CMTR1–DHX15 interaction was not having a broad impact on translational control (Fig 7C). To investigate gene-specific effects, RNA sequencing (RNAseq) analysis was used to quantify total cellular transcripts and the transcripts associated with most dense ribosome binding (Fig 7A, shaded area, and Tables S1 and S2). Out of the 12,238 gene transcripts that passed quality thresholds, none exhibited a significant difference in expression level in cells expressing CMTR1 WT and 2L/A (Table S1 and Fig 7D). This indicates that in HCC1806 cells, the CMTR1–DHX15 interaction is unlikely to have a significant impact on

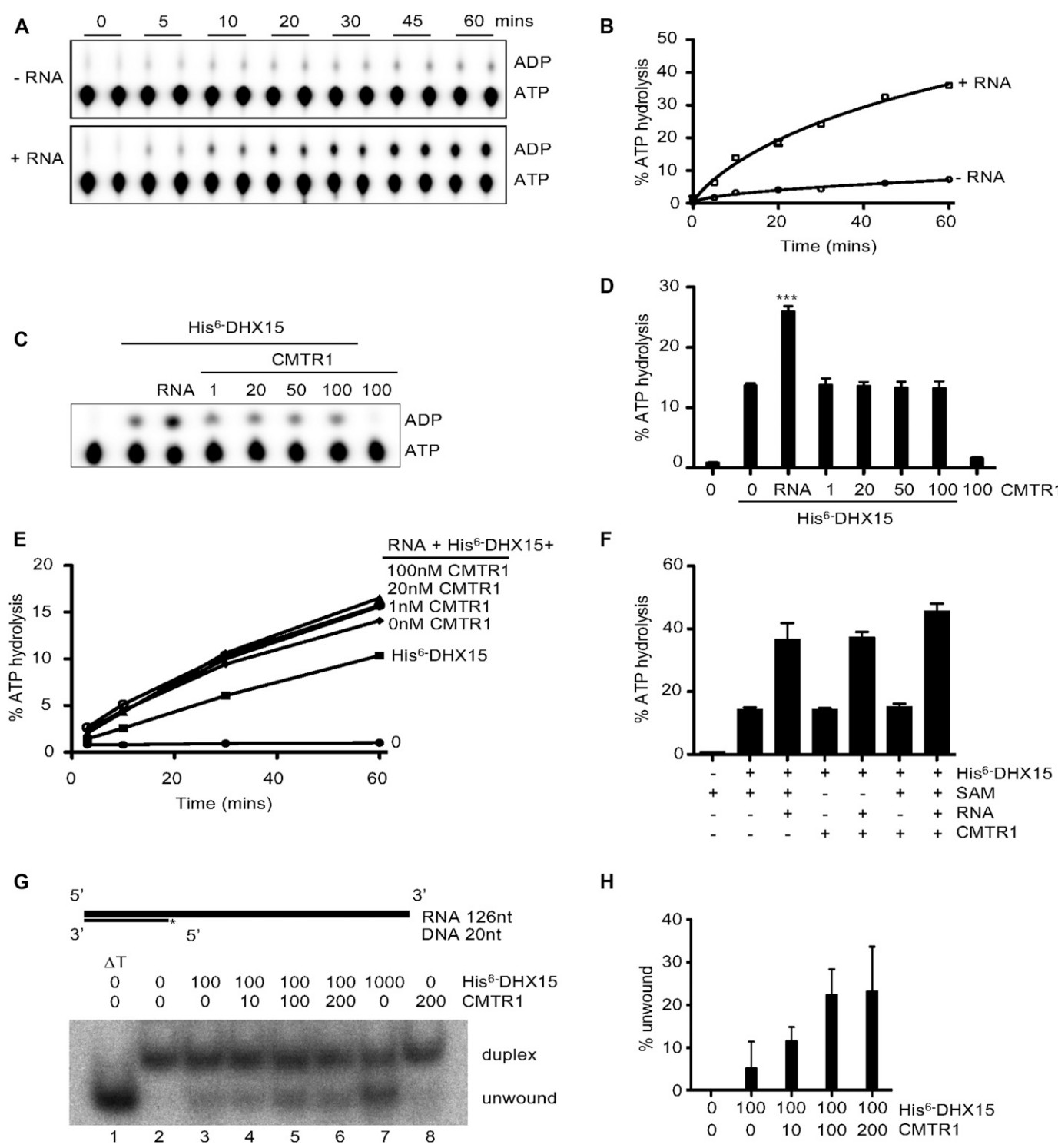

**Figure 4. CMTR1 methyltransferase stimulates DHX15 helicase activity.**
**(A)** 100 nM His[6]-DHX15 was incubated with $\alpha^{32}$P-ATP in the absence or presence of 1 μg HeLa cell RNA for the time indicated. The ADP generated was resolved by TLC and detected by phosphoimager throughout the figure unless indicated. **(B)** Average percentage of ATP hydrolysis and standard deviation are plotted for two experiments. **(C)** 100 nM His[6]-DHX15 was incubated with $\alpha^{32}$P-ATP and 1 μg RNA or nM CMTR1 indicated for 30 min. ATP hydrolysis was detected. **(D)** Average percentage ATP hydrolysis and standard deviation are plotted for three independent experiments. t test was performed relative to DHX15 alone. ***P-value < 0.005. **(E)** 100 nM His[6]-DHX15 alone or 100 nM His[6]-DHX15, 1 μg HeLa RNA, and indicated concentration of CMTR1 were incubated with $\alpha^{32}$P-ATP for 30 min. Percentage of ATP hydrolysis is plotted over time. **(F)** 100 nM DHX15 was incubated with 10 μM SAM, 1 μg HeLa RNA, and 100 nM CMTR1 as indicated for 30 min. Percentage of ATP hydrolysis is plotted. **(G)** RNA-$^{32}$P-DNA duplex was incubated with with1mM ATP and indicated nM His[6]-DHX15 and CMTR1. After 30 min, samples were resolved by native PAGE and visualized using a phosphorimager. "Unwound" indicates single stranded $^{32}$P-DNA oligonucleotide. ΔT is the 95°C denatured substrate. **(H)** Average percentages of unwound substrate and standard deviations are presented for three independent experiments.

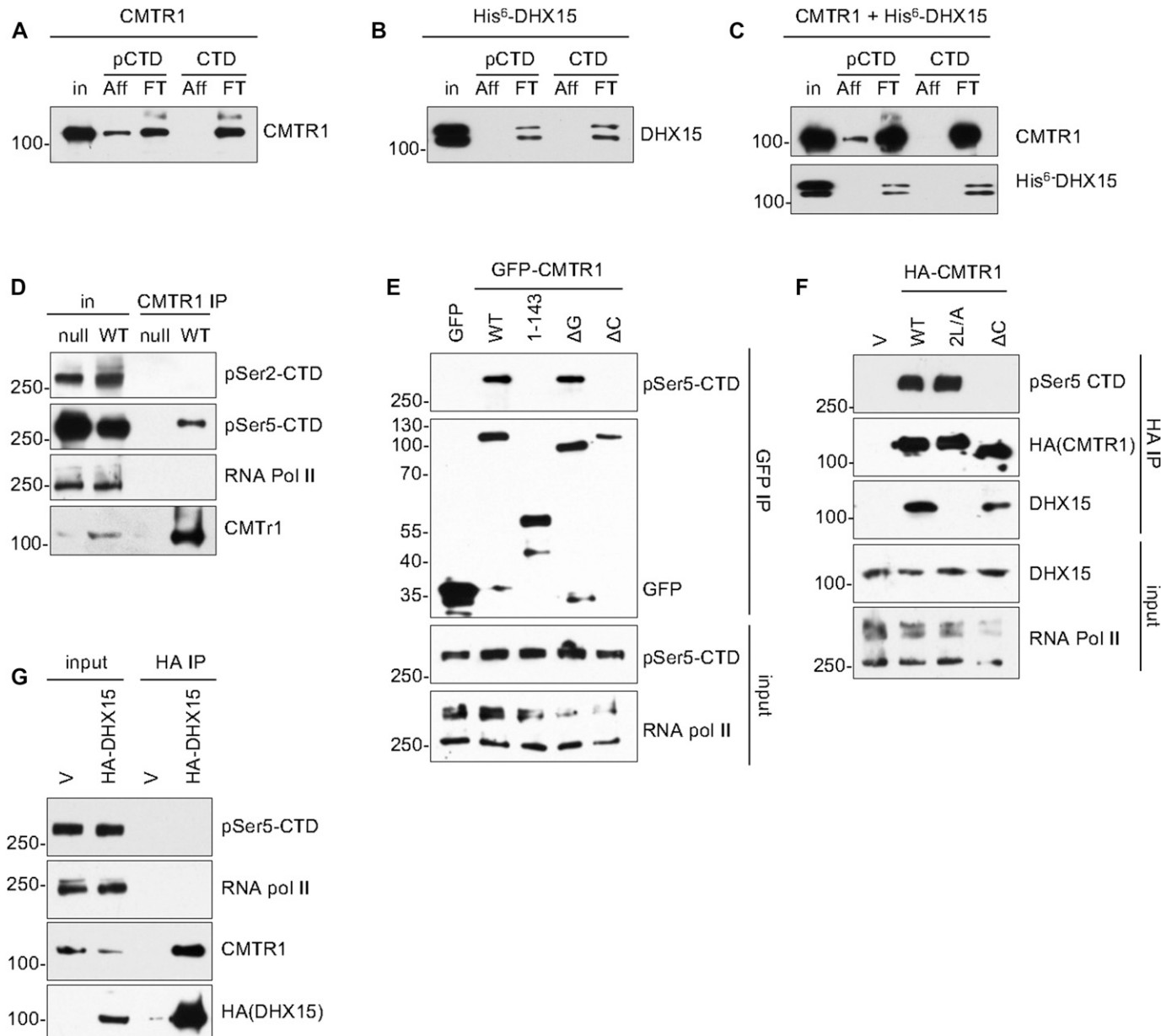

**Figure 5. CMTR1 interaction with DHX15 and RNA pol II CTD are mutually exclusive.**
**(A)** Recombinant CMTR1, **(B)** recombinant His⁶-DHX15, **(C)** pre-mixed CMTR1, and His⁶-DHX15, were incubated with biointinylated peptides of three RNA pol II CTD heptad repeats, either unphosphorylated (CTD) or phosphorylated on S5 (pCTD). Peptides were enriched on streptavidin beads and associated proteins analysed by Western blot. **(D)** CMTR1 was immunoprecipitated from extracts of HeLa cells (WT) or HeLa *CMTR1* null cells. Western blot analysis was performed. **(E)** pcDNA5 GFP-CMTR1 or indicated mutants were expressed in HeLa cells. IPs were performed with anti-GFP antibodies and analysed by Western blot. **(F)** pcDNA 5 HA-CMTR1 or indicated mutants were expressed in HeLa cells. IPs were performed with anti-HA antibodies and analysed by Western blot. **(G)** pcDNA5 HA-CMTR1 and mutants were expressed in HeLa cells. IPs were performed with anti-HA antibodies and analysed by Western blot. Aff, affinity purified fraction; In, input; FT, flow through.

transcription or RNA stability. However, 59 gene transcripts were significantly enriched in polysomes in cells expressing CMTR1 2L/A compared with WT. This indicates that the DHX15–CMTR1 interaction restricts the translation of a subset of mRNAs (Fig 7E and F and Table S2). Conversely, no genes were significantly depleted from polysomes in cells expressing CMTR1 2L/A compared with WT.

The genes that exhibited enhanced translation in cells expressing CMTR1 2L/A were enriched for gene ontology terms associated with important metabolic functions and cell cycle processes (Table S3). Notably, there was enrichment for gene encoding factors involved with the cell cycle and DNA damage responses including *ATR*, *UBR4*, *UBR2*, *CENPE*, *PRKDC*, and *BIRC6*; factors involved in RNA processing including *GCN1*, *TPR*, *RANBP2*, and *NUP205*; metabolic enzymes including *FASN*, *EPRS*, and *CAD*; and focal adhesion-associated molecules *HSPG2*, *FLNB*, *MYH9*, *FAT1*, *IGF2R*, and *CLTC*. The polysome loading of selected genes was confirmed by RT–PCR and enhanced protein expression confirmed by Western blot (Fig 7G and H). This analysis indicates that DHX15

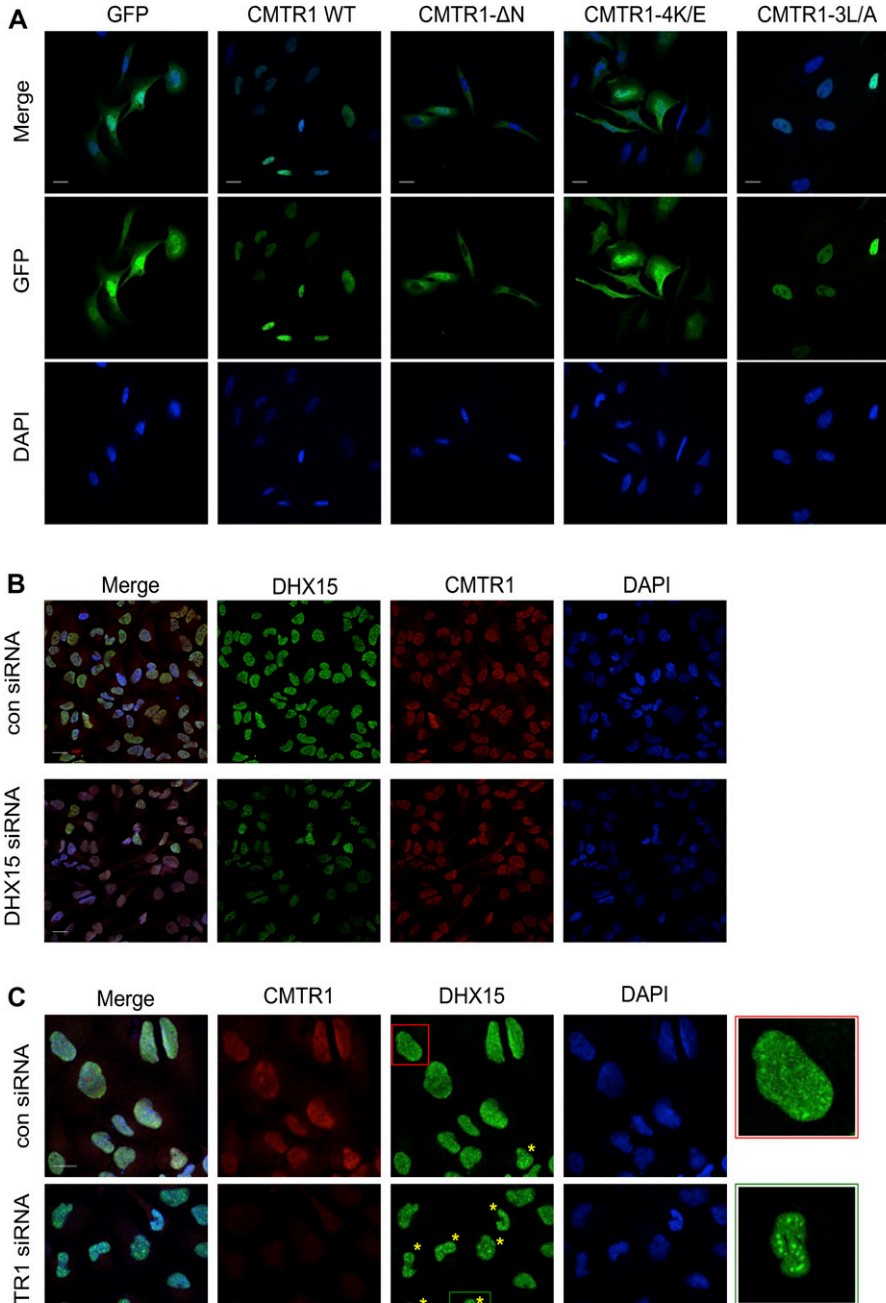

**Figure 6. DHX15 localisation is influenced by CMTR1.**
**(A)** Representative fluorescence images of GFP-CMTR1 WT or mutants expressed in HeLa cells. Cells were DAPI-stained to visualise DNA. **(B)** HeLa cells were transfected with DHX15 siRNA or non-targeting control. Representative immunofluorescence (IF) images of endogenous CMTR1 and DHX15 presented. Images show CMTR1 (red), DHX15 (green), and DAPI (blue). **(C)** As in (B), except cells were transfected with CMTR1 siRNA or control. Yellow asterisks indicate cells with DHX15 accumulation in foci. 4.5× magnified areas marked. Data representative of three independent experiments. Bar indicates 20 μm.

represses the positive effect of CMTR1 on cell growth and proliferation.

## The interaction of DHX15 and CMTR1 inhibits proliferation

The impact of CMTR1 expression on cell proliferation was investigated in HCC1806 cells and another mammary epithelial tumour line, MCF7. Transfection of two independent CMTR1 siRNAs resulted in the suppression of CMTR1 expression and a reduction in cell proliferation (Fig 8A–C). Given that expression of CMTR1 2L/A

resulted in increased polysome loading of genes involved in metabolism and cell cycle, we investigated its impact on cell proliferation. When HA-CMTR1 was transiently expressed in sparsely plated HCC1806 cells, a significant increase in cell number was observed after 24 h (Fig 8D). Transient expression of HA-CMTR1 2L/A resulted in a further increase in cell number, supporting the hypothesis that DHX15 suppresses the translation of a subset of progrowth genes. To investigate if the increased expression of these genes may contribute to the enhanced proliferation observed on expression of CMTR1 2L/A, two genes with increased polysome

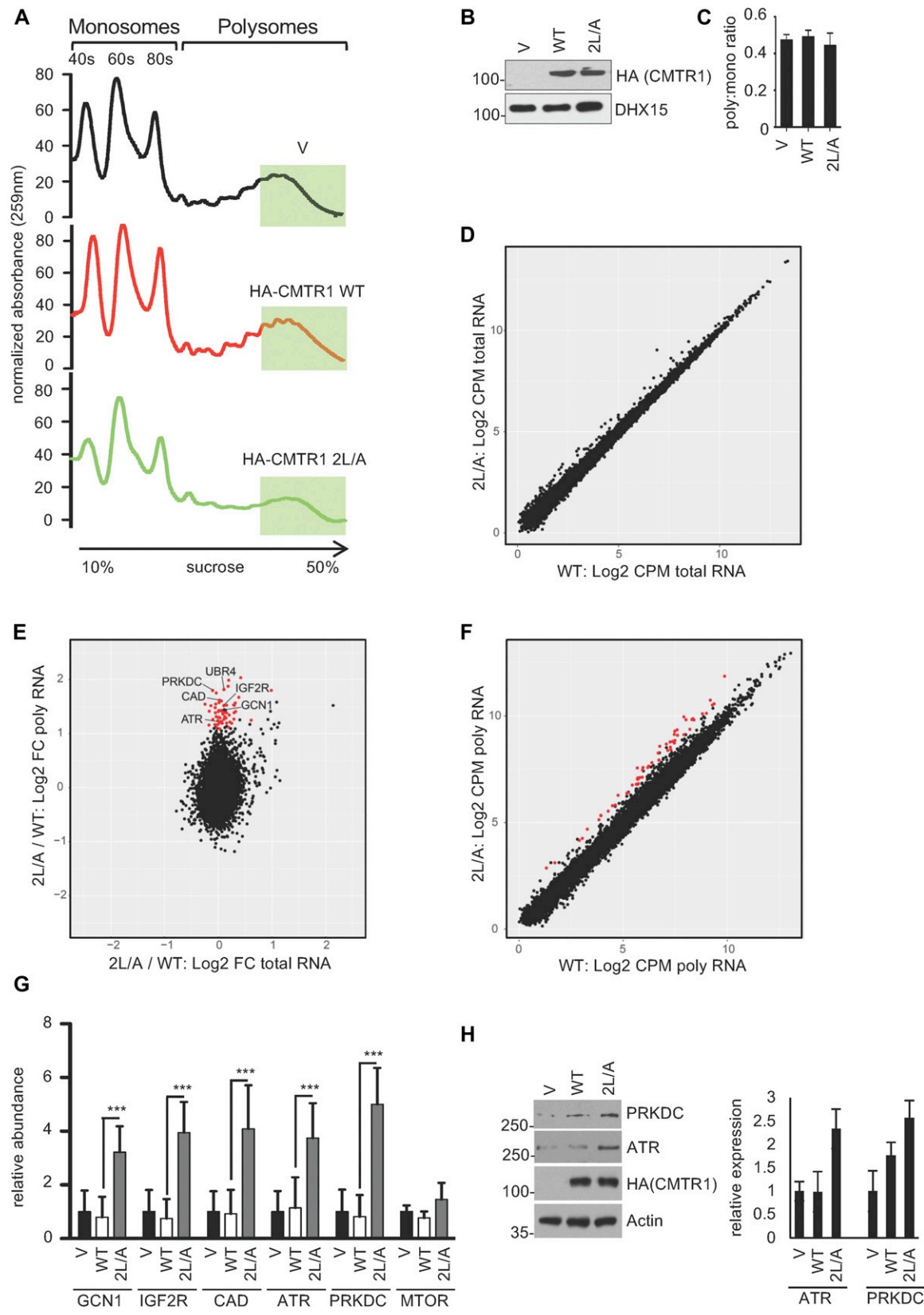

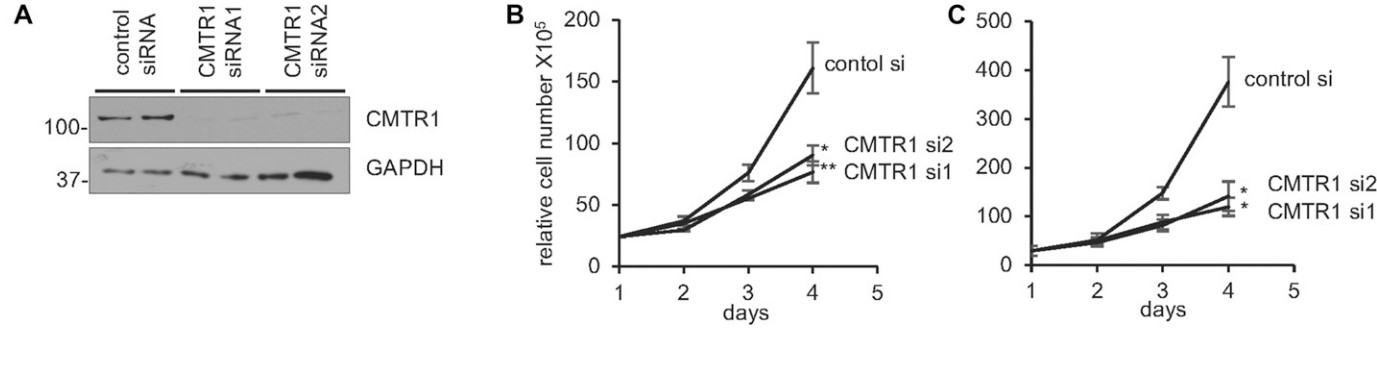

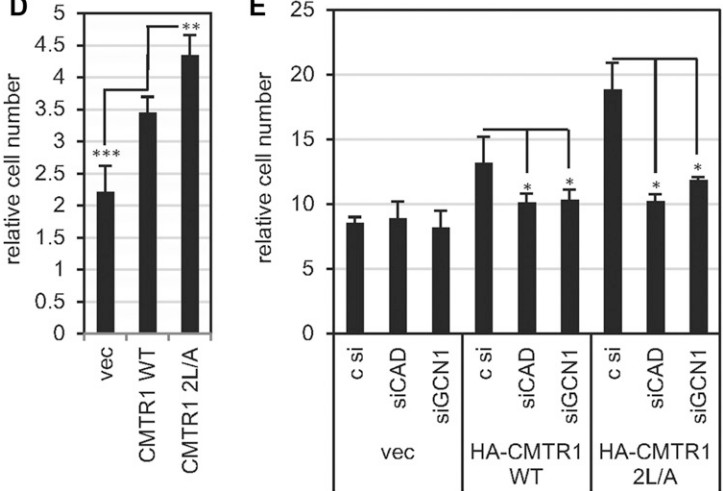

**Figure 8. The DHX15–CMTR1 interaction inhibits cell proliferation.**
**(A)** MCF7 cells were transfected with two independent CMTR1 siRNAs or non-targeting control. Cell extracts were analysed by Western blot. **(B)** MCF7, **(C)** HCC1806 cells were transfected with two independent CMTR1 siRNAs or non-targeting control. Cells were counted every day. HCC1806 were transfected with **(D)** pcDNA5 CMTR1 WT or 2L/A or vector control. After 48 h cells were counted. Average and standard deviation presented for 3–5 independent wells. **(E)** As in (D) except cells were also transfected with and 50 nM CAD, GCN1 or non-targeting control siRNA. *t* test was performed. *P-value < 0.05; **P-value < 0.01; ***P-value < 0.005.

loading in CMTR1 2L/A cells, CAD and GCN1, were suppressed by siRNA transfection (Fig 8E). Suppression of CAD and GCN1 resulted in reduced proliferation of HCC1806 cells, supporting the hypothesis that DHX15 controls CMTR1-dependent translation of a subset of pro-growth genes.

## Discussion

Higher eukaryotes carry unique mRNA cap modifications, including first transcribed nucleotide ribose O-2 methylation, which is associated with translation and self-RNA identification (Kuge et al, 1998; Schuberth-Wagner et al, 2015; Leung & Amarasinghe, 2016). We investigated cellular regulation of CMTR1, the first nucleotide ribose O-2 methyltransferase (Belanger et al, 2010). CMTR1 was isolated from mammalian cells in a complex with DHX15 (human orthologue of yeast Prp43), a DEAH-box helicase involved in RNA processing, splicing, and ribosome biogenesis (Koodathingal & Staley, 2013; Robert-Paganin et al, 2015). DHX15 has an OB-fold domain through which it binds to the G-patch domain in a series of proteins, facilitating contribution to several nuclear functions (Niu et al, 2012; Chen et al, 2014; Robert-Paganin et al, 2015; Memet et al, 2017; Tauchert et al, 2017). Previous studies indicated that CMTR1 and DHX15 participate in the same mRNA processing events (Yoshimoto

**Figure 7. The DHX15–CMTR1 interaction inhibits translation of a subset of genes.**
**(A)** Extracts from HCC1806 cells transfected with pCDNA5 HA-CMTR1, 2L/A, or vector control (V) were centrifuged through 10%–50% sucrose gradients. 259 nm absorbance was determined across the gradient in four independent experiments. Representative experiment presented. The fraction of the gradients analysed by RNAseq is indicated by shaded boxes. The identities of the peaks in the polysome profiles is indicated. **(B)** Cell extracts were analysed by Western blot. **(C)** Average and standard deviation of ratio of polysome to mononosome absorbance for four independent experiments. RNAseq analysis was used to determine uniquely aligned reads per gene (transcript isoforms collapsed) in RNA and polysome samples from HCC1806 cells expressing HA-CMTR1 WT and 2L/A (n = 2). **(D)** Scatter plot of log2 transformed total counts per million (log2CPM) in total RNA. **(E)** Scatter plot of log2-fold change (FC) in 2L/A/WT for total RNA and polysomal RNA. Significantly regulated genes determined by a negative binomial rest in EdgeR in red. **(F)** Scatter plot of log2-transformed counts per million (log2CPM) in polysomal RNA. **(G)** Polysomal mRNA levels quantified by RT–PCR. Average and standard deviation for three independent experiments. *t* test was performed ***P-value < 0.005. **(H)** Western blot analysis of proteins in HCC1806 cells expressing pCDNA5 HA-CMTR1, 2L/A, or vector control (V). Charts represented average protein signal in Western blots and standard deviation for three independent experiments. Quantitation performed in ImageJ software.

et al, 2014; Gebhardt et al, 2015). Here, we demonstrate that the direct interaction of CMTR1 and DHX15 regulates the catalytic activity of both enzymes, which impacts gene expression and cell proliferation.

### DHX15 constrains CMTR1 function

mRNA cap formation initiates early during transcription, when RNGTT (RNA triphosphatase/guanylyltransferase) and RNMT (N-7 cap guanosine methyltransferase) are recruited to Ser-5 phosphorylated RNA pol II CTD (Ramanathan et al, 2016). We report that CMTR1 also interacts directly with Ser-5 phosphorylated RNA pol II CTD. DHX15 reduces CMTR1 methyltransferase activity in a dose-dependent manner and DHX15–CMTR1 complexes are not recruited to RNA pol II. Thus, the interaction of DHX15 with CMTR1 is likely to constrain first nucleotide O-2 methylation to the initial stages of transcription, and restrain post-transcriptional, aberrant methylation. We demonstrate that CMTR1 interaction with DHX15 requires the G-patch domain, whereas the CMTR1 interaction with RNA pol II CTD requires the WW domain. The CMTR1-DHX15 complex may be unable to be recruited to RNA pol II either because DHX15 changes the conformation of CMTR1 or sterically hinders it from binding to the CTD.

To our knowledge, CMTR1 is the only annotated G-patch–containing protein with a catalytic domain and this is the first example of DHX15 regulating a catalytic activity. Interaction with DHX15 is required for inhibition of CMTR1 methyltransferase activity; however, the helicase/ATPase activity of DHX15 is not. The catalytic domain lies in the centre of the CMTR1 polypeptide and the non-catalytic C- and N-terminal domains influence methyltransferase activity (Smietanski et al, 2014). Our data are consistent with DHX15 inducing a conformational change in CMTR1 to reduce methyltransferase activity.

First nucleotide ribose O-2 methylation is associated with translational efficiency (Muthukrishnan et al, 1976a; Kuge & Richter, 1995; Kuge et al, 1998). The impact of DHX15 on CMTR1-dependent translation was investigated using the CMTR1 2L/A mutant, which does not bind to DHX15 but is recruited to RNA pol II. Expression of CMTR1 2L/A resulted in increased ribosome loading of a subset of transcripts, including those involved in metabolic pathways and cell cycle control. The genes sensitive to the DHX15–CMTR1 interaction may be particularly dependent on O-2 methylation for polysome loading or require high levels of CMTR1 activity to be O-2 methylated. Expression of CMTR1 2L/A also resulted in increased cell proliferation.

### CMTR1 influences DHX15 function

The mechanism by which G-patch proteins regulate the activity and localisation of Prp43 (yeast DHX15 orthologue), are well characterised. Prp43 uses the energy generated by ATP hydrolysis to power helicase activity (Walbott et al, 2010). The Prp43 RecA domains interact with the C-terminal domain, rendering the enzyme in a closed conformation. Disruption of these interactions by ATP binding promotes changes in the helicase structure, leading to the open conformation required for unwinding complex RNA stretches (Tauchert et al, 2017). In addition, the stacking of the adenosine base with the RecA1 R and RecA2 F motifs is important for the activity and regulation of the helicase (Robert-Paganin et al, 2017). When intrinsically unstructured G-patch domains bind to the Prp43

OB-fold, they adopt a stable open secondary structure (Christian et al, 2014). G-patch domains can influence both the stacking of the adenosine base and interactions of N- and C-terminal domains, thus regulating Prp43 catalytic activity (Robert-Paganin et al, 2017; Tauchert et al, 2017).

We observe that CMTR1 activates DHX15 helicase activity. Similar to several other G-patch proteins, CMTR1 activates helicase activity without regulating ATPase activity (Memet et al, 2017). A similar observation was made with the yeast splicing factor Ntr1, which does not alter Prp43 ATPase activity but does activate the helicase (Tanaka et al, 2007). Furthermore, there are now several examples of G-patch proteins impacting differentially on Prp43 activity and function (Aravind & Koonin, 1999; Banerjee et al, 2015; Tauchert et al, 2017).

The competition of cofactors for Prp43 was previously observed to regulate its distribution between different pathways (Heininger et al, 2016). Consistent with a previous publication, we observed DHX15 co-localising with a splicing factor, SC35, in nuclear speckles (Tannukit et al, 2009). Suppression of CMTR1 expression resulted in an increased number of cells with DHX15 in nuclear speckles. CMTR1 may compete with other G-patch proteins for DHX15 binding and/or the impact that CMTR1 has on the cell cycle may indirectly influence the number of nuclear speckles. The indirect impact of CMTR1 suppression can be observed by a change in nuclear morphology from spheroid to lobed (Fig 6), and by an impact on cell proliferation (Fig 8).

### The co-ordinated impact of DHX15 and CMTR1 on gene expression

Ultimately, we want to understand how the relationship between DHX15 and CMTR1 impacts on cell function. This is complex because of the multifunctional nature of both enzymes. The impact of the DHX15–CMTR1 interaction will depend on the relative expression of the enzymes and their other interacting partners, which also influence enzyme activity and localisation. The impact of the DHX15–CMTR1 interaction will also depend on the underlying cell physiology, including relative dependency on O-2 methylation, splicing, rRNA processing, and translation. Of note, CMTR1 is an interferon-regulated gene, whereas DHX15 is not, and therefore, interferon signalling alters the ratio of these factors. In this study, we focus on the biological impact of DHX15 repression of CMTR1 methyltransferase activity and function. We stress that, in certain contexts, the impact of CMTR1 activation of DHX15 helicase activity may be equivalently or more biologically important, given the role of DHX15 in RNA processing.

In summary, we now recognise that the mRNA capping enzymes, RNGTT, RNMT, and CMTR1 are regulated by different co-factors and posttranslational modifications, with multiple impacts on gene expression and cell physiology. The challenge now is to understand the complex integration of these regulatory events during development and in the adult.

## Materials and Methods

### Cell culture

HeLa, HEK293, MCF7, and U2OS cells were cultured in DMEM and HCC1806 cells were cultured in Roswell Park Memorial Institute 1640

medium. All cells were cultured with 10% (vol/vol) FBS, 100 U/ml penicillin, 0.1 mg/ml streptomycin, and 2 mM ʟ-glutamine. For plasmid transfection, cells in 10-cm dishes were transfected with 1–5 μg plasmid using 20 μg polyethylenimine (Polysciences). Cells were cultured for 36 h before lysis in 0.45 ml ice-cold lysis buffer (50 mM Tris–HCl, pH 7.5, 1 mM EGTA, 1 mM EDTA, 1 mM sodium orthovanadate, 10 mM β-glycerophosphate, 50 mM NaF, 5 mM sodium pyrophosphate, and 0.27 M sucrose) supplemented with 1% (vol/vol) Triton X-100, 2 mM DTT, 1% (vol/vol) aprotinin, 10 μM leupeptin, and 1 μM pepstatin. Extracts were centrifuged at 16,200 g at 4°C for 15 min and supernatant retained. For siRNA transfections, cells were transfected with 50 nM siRNA (Dharmacon siGenome range; non-targeting or CMTR1), for 48 h unless stated otherwise. UAGAUGAUGUUCGGGAUUA, 01 CMTR1 siRNA; GUAAGAGCGUGUUU-GAUGU, 02 CMTR1 siRNA. For plasmid and siRNA co-transfection, lipofectamine 2000 (Thermo Fisher Scientific) was used. 2–6 × 10⁴ cells were plated in a six-well plate, 0.5 μg pcDNA5 GFP, and 0.5 μg pcDNA5 HA-CMTR1 WT or 2L/A mixed with siRNA before transfection. Countess cell counter was used for cell counting.

## Generation of a CMTR1 knockout cell line (CMTR1 del) using CRISPR/Cas9

CMTR1 gene (ensembl ENSG00000137200) optimal scoring guide target site GGGAGGTTCATCATCGGACG[TGG] was cloned into pBabeD pU6 and sequence-verified (http://crispr.mit.edu/). HeLa cells stably expressing Tet-regulated Cas9 were transfected with 3 μg pBabeD pU6 CMTR1 using Lipofectamine 2000. Cells were incubated in DMEM, 10% FBS, 2 mM ʟ-glutamine, and 100 μg/ml Normocin (InvivoGene). After 12 h, 4 μg/ml puromycin was added in fresh medium. After 48-h selection, 2 μg/ml doxycycline was added to induce Cas9 expression. After 72 h, single cells were FACS-sorted into a 96-well plate containing DMEM/20% FBS, 2 mM L-glutamine, 100 U/ml penicillin, 100 μg/ml streptomycin, and 100 μg/ml Normocin (InvivoGene). CMTR1 protein expression was screened using Western blot in 40 clones. For clones displaying an absence of protein, genomic DNA was isolated and amplified by PCR, using *CMTR1* genomic forward: CTGTGATCCCAGTGGCTGT and *CMTR1* genomic reverse: CCAAGGGGCAGTGGACTAT primers. PCR products from control and *CMTR1* del clones were sequenced. *CMTR1* del clone 5 had no region homology to WT cells around Cas9 targeting sequence and selected for experiments.

## IP and Western blot

For IP of GFP or HA-tagged proteins, anti–GFP antibody–conjugated agarose (ChromoTek) and anti-HA antibody–conjugated agarose (Sigma) were used. For endogenous proteins, 1 μg relevant antibody was pre-incubated with 10 μl protein-G Sepharose packed beads and washed to remove non-bound antibody. 0.5–2.5 mg lysates were precleared with 10–30 μl protein-G Sepharose (GE Healthcare), and then incubated with an antibody–resin conjugate for 2 h at 4°C under gentle agitation. IPs were washed three times with lysis buffer containing 0.1 M NaCl. Typically, 1%–2% input and 30% of IP eluates were loaded for Western blot, except in Figs 1F and 4A–C in which 15% of eluate was loaded. For RNAse treatment, 40 μg RNAse A was incubated with IP or 2 μg HeLa RNA for 60 min at 4°C. Proteins

were eluted in SDS-sample buffer. Western blots were performed by standard protocols. CMTR1 antibody was raised in sheep by Orygen Antibodies Limited, UK and affinity purified against the human recombinant protein. Second bleed was used at 1 μg/ml for Western blotting and the first bleed was used at 1 μg/IP. Other antibodies: RNGTT (in-house), HA (Sigma), DHX15 (ab70454; Abcam), GAPDH (Cell Signaling Technologies), RNA pol II (N20; Santa Cruz), PSer5-PolII (3E8; ChromoTek), and pSer2-PolII (3E10; ChromoTek). Secondary antibodies were from Pierce.

## Mass spectrometry analysis

IPs were resolved using SDS–PAGE and stained with Novex Colloidal Blue (Invitrogen). Bands were excised and washed sequentially on a vibrating platform with 0.5 ml water, 1:1 (vol/vol) water and acetonitrile (AcN), 0.1 M ammonium bicarbonate, and 1:1 (vol/vol) 0.1 M ammonium bicarbonate and AcN. Samples were reduced in 10 mM dithiothreitol (20 min, 37°C), alkylated in 50 mM iodoa-cetamide/0.1 M ammonium bicarbonate (20 min, dark), and washed in 50 mM ammonium bicarbonate and 50 mM ammonium bicarbonate/50% AcN. When gel pieces were colourless they were washed with AcN for 15 min and dried. Gel pieces were incubated (16 h, 30°C) in 5 μg/ml trypsin/25 mM triethylammonium bicarbon-ate. Tryptic digests were analysed by liquid chromatography–mass spectrometry on an Applied Biosystems 4000 QTRAP system with precursor ion scanning. The resulting MS/MS data were analysed by Mascot search algorithm (http://www.matrixscience.com).

## Bacterial expression and purification of DHX15 and CMTR1 proteins

Bacterial expression plasmids pET15b-His⁶-DHX15 WT or E261Q and pGEX6P1-GST-3C-CMTR1 WT or ΔG (residues 143–835) were trans-formed into Rosetta 2 (DE3) cells (Novagen) and plated on LB agar/100 μg/ml ampicillin and 35 μg/ml chloramphenicol. Starter cul-tures were incubated at 37°C overnight. After 16 h, cultures were diluted 1/50 into fresh 6-9L LB cultures and incubated at 37°C with agitation. At OD600 ~ 0.3–0.4, the temperature was reduced to 16°C and at OD₆₀₀ ~ 0.7–0.8 protein expression was induced with 50 μM Isopropyl β-D-1-thiogalactopyranoside for 14–16 h. Cultures were harvested by centrifugation and pellets resuspended in ice-cold re-suspension buffer (Tris 50 mM, pH 8, 200 mM NaCl, 1 mM DTT, 25 mM imidazole, 10% glycerol [vol/vol], and Complete-EDTA free protease inhibitor purchased from Roche). Resuspended cells were flash-frozen in liquid nitrogen, thawed in a lukewarm (25°C) water bath, and treated with lysozyme (1 mg/ml) and DNase I (1:10,000 dilution) for 30 min at 4°C. Cell debris and insoluble material were clarified by centrifugation at 38,000 relative centrifugal force at 4°C for 45 min. Soluble lysates were passed through poly-prep columns packed with either Ni-NTA agarose (His⁶-DHX15) or GSH-sepharose (GST-3C-CMTR1) pre-equilibrated in wash buffer (50 mM Tris, pH 8, 500 mM NaCl, 1 mM DTT, 25 mM imidazole, and 5% glycerol [vol/vol]). After binding, resins were extensively washed with wash buffer followed by a final wash step in low-salt (200 mM NaCl) wash buffer. For His⁶-DHX15, Ni-NTA bound material was eluted with an elution buffer (50 mM Tris, pH 8, 200 mM NaCl, 1 mM DTT, 5% glycerol, and 300 mM imidazole), whereas GSH-sepharose–bound GST-CMTR1

was treated with GST-3C protease overnight at 4°C to cleave GST tag. Eluted His$^6$-DHX15 and CMTR1 proteins were concentrated using an Amicon 30 kD- molecular weight cut off concentrator and subjected to size-exclusion chromatography (Superdex 200) in 50 mM Tris, pH 8, 200 mM NaCl, 1 mM DTT, and 5% glycerol (vol/vol) buffer on an AKTA purifier FPLC system (Amersham). Proteins were further concentrated and flash-frozen as single use aliquots, stored at –80°C.

## Gel filtration

0.5 ml of 2 mg/ml cell extract was 0.22-$\mu$m filtered and resolved on Superdex 200 10/300 GL preparative grade column (GE Healthcare) in 50 mM Tris–HCl, pH 7.4, 1 mM EDTA, 0.1 M sodium chloride, 0.03% Brij-35, and 2 mM DTT at 0.4 ml/min flow. 0.5 ml fractions were collected. Molecular weight markers (Bio-Rad) were used: bovine thyroglobulin (670 kD), bovine gamma globulin (158 kD), and chicken ovalbumin (44 kD).

## First nucleotide O-2 methylation assay

A guanosine-capped substrate $^{32}$P-labelled on $\alpha$-phosphate (GpppG-RNA) was produced. 200 ng 55-base in vitro–transcribed RNA was incubated with 100 ng RNA guanylyltransferase and 5′-phosphatase (RNGTT), 2 $\mu$l (10 $\mu$Ci) [$\alpha^{32}$P]GTP, and 1 $\mu$l RNAsin (Promega) in 10 $\mu$l reaction buffer (0.05 M Tris–HCl, pH 8.0, 6 mM KCl, and 1.25 mM MgCl$_2$) at 37°C for 60 min. When used, a guanosine-capped transcript was incubated with 100 ng RNMT and 0.2 $\mu$M SAM for 20 min at 30°C to produce N-7 methylated guanosine cap (m7GpppG-RNA). RNA was purified by ammonium acetate precipitation. Methyltransferase assay performed in 10 $\mu$l reaction buffer, with 3 nM CMTR1 (or indicated nM), 2 ng $^{32}$P-labeled GpppG-RNA or m7GpppG-RNA, and 10 $\mu$M SAM at 30°C for 30 min (unless otherwise stated). Following the reaction, RNA was precipitated and resuspended in 4 $\mu$l 50-mM sodium acetate and 0.25 U P1 nuclease for 30 min at 37°C. Cap structures were resolved in 0.4 M ammonium sulphate on polyethyleneimine–cellulose plates, and visualized and quantified by autoradiography. In Fig 3E, when cell extracts were used in the assay, cells were lysed in a lysis buffer (as above) and 5 $\mu$g cell extract was used in the methyltransferase reaction.

## Molecular biology

cDNA cloning and mutagenesis were performed by standard protocols. Constructs were sequence-verified.

## CTD peptide affinity chromatography

RNA pol II CTD peptide chromatography was performed as in Ho & Shuman (1999). 0.5 nmol biotinylated CTD peptides were absorbed on 0.2 mg Streptavidin Dynabeads M-280 (Invitrogen) in 300 $\mu$l binding buffer (25 mM Tris–HCl, pH 8, 50 mM NaCl, 1 mM DTT, 5% Glycerol, and 0.03% Triton X-100). After 45 min at 4°C, beads were magnet-concentrated and washed in binding buffer. 0.2 $\mu$g indicated protein was mixed with peptide-beads in 50 $\mu$l binding buffer. After incubation for 45 min at 4°C, beads were washed three

times with a binding buffer and bound fraction eluted with 30 $\mu$l SDS–PAGE loading buffer at 95°C for 5 min.

## ATPase activity assay

ATPase reactions performed as in Lebaron et al (2009). 100 nM His$^6$-DHX15, 0.6 $\mu$Ci/$\mu$l [$\alpha^{32}$P] ATP, and 100–200 $\mu$M ATP were incubated in 5 $\mu$l buffer (25 mM Tris–acetate, pH 8.0, 2.5 mM Mg(CH3-COO)$_2$, 100 mM KCl, 0.2 mM DTT, 100 $\mu$g/ml BSA [Sigma]) at 30°C for the time indicated. 1 $\mu$l reaction mix was resolved on polyethyleneimin–cellulose plates using 0.75 M KH$_2$PO$_4$. Spots were visualised and quantified on a phosphorimager. When indicated, 1 $\mu$g HeLa cell RNA was added to the reaction.

## Cap-binding assays

Cap-binding assays were performed as in Rio (2014) with minor modifications. Reactions were performed with $^{32}$P-labeled GpppG-RNA as in O-2 methylation assays with indicated amount of enzymes. After 20 min incubation, reactions were stopped with 5 $\mu$l of loading buffer (10 mM Tris, pH 7.5, 60 mM KCl, 5% glycerol, 0.01% xylene cyanol, and bromophenol blue), loaded, and resolved in 4%–12% tris borate EDTA buffer non-denaturing acrylamide gel (Invitrogen) for 75 min at constant 50 V and 4°C. Gel were fixed (10% acetic acid and 10% methanol) for 5 min, washed in 10% glycerol for 10 min, and detected using a phosphorimager.

## Unwinding assay

Unwinding assay was performed as in Tanaka et al (2007). The 126-nucleotide (nt) RNA strand (5′-GGGCGAAUUGGGCCCGACGUCGC-AUGCUCCCGGCCGCCAUGGCGGCCGCGGGAAUUCGAUUAUCACUAGU-GAAUUCGCGGCCGCCUGCAGGUCGACCAUAUGGGAGAGCUCCCAACGC-GUUGGAUG-3′) was synthesized by in vitro transcription, and annealed at a 1:3 M ratio to 5′ $^{32}$P-labelled DNA (5′-GACGTCGGGCCCAATTCGCCC-3′) to yield 5′-tailed 30-bp duplex. The annealed substrate was gel-purified on native 6% PAGE. 10 $\mu$l reaction mixtures containing indicated concentrations of His$^6$-DHX15 and CMTR1, and 2.5 nM RNA/DNA substrate were incubated in 40 mM Tris–HCl, pH 7.0, 2 mM DTT, and 1 mM ATP-Mg$^{2+}$ at 37°C for 45 min. Reactions were halted by transfer to ice and addition of 5 $\mu$l loading buffer (100 mM Tris–HCl, pH 7.4, 5 mM EDTA, 0.5% SDS, 50% glycerol, 0.1% [wt/vol] bromophenol blue, and xylene cyanol dyes). Samples were resolved on 16% polyacrylamide gel in 40 mM Tris–borate, 0.5 mM EDTA, and 0.1% SDS. $^{32}$P-labeled substrate and products were visualized by autoradiography.

## Immunofluorescence

Immunofluorescence was performed in 2% BSA/PBS at room temperature. Cells were fixed in 4% paraformaldehyde for 10 min, permeabilized with 1% NP-40/PBS for 3 min, blocked with 10% donkey serum for 20 min, and incubated in 1.4 $\mu$g/ml polyclonal sheep anti-CMTR1 or DHX15 antibodies for 1.5 h, washed, and incubated with 1.4 $\mu$g/ml Alexa Fluor 594– or 488–conjugated donkey anti-sheep or rabbit antibodies for

45 min. Cells were counterstained with 1 µg/ml DAPI, mounted in 2.5% 1, 4-diazobicyclo-(2,2,2-octane), and visualized by fluorescence microscopy (LSM 700; Zeiss).

### Polysome profile

Cells were incubated in 100 µg/ml cycloheximide for 10 min and washed in ice-cold PBS supplemented with 100 µg/ml cycloheximide, and extracts were collected in polysome lysis buffer (15 mM Tris, pH 7.5, 15 mM MgCl$_2$, 0.3 M NaCl, 1 mM DTT, 1% Triton X-100, 100 mg/ml cycloheximide, and 100 U/ml RNasin). 10% extracts were retained as input and 90% resolved by centrifugation through 10 ml 10%–50% sucrose gradient at 18,000 $g$ for 2 h at 4°C. Fractions were collected on a FoxyR1 fractionator (Teledyne ISCO) with OD259 nm monitoring.

### RNA-sequencing and analysis

Extraction of polysomal RNA was performed as described previously (Grasso et al, 2016). Briefly, polysomal fractions 16–20 were purified using Phenol:Chloroform:Isoamyl alcohol (25:24:1). RNA was precipitated overnight with 2 M of LiCl, 20 mM Tris, pH 7.5, and EDTA 10 mM. Input RNA was purified using Trizol Reagent. RNA was sequenced at the Tayside Centre for Genomic Analysis. RNA was quality checked using TapeStation (Agilent Technologies). RNAseq libraries were prepared with TruSeq Stranded Total RNA with Ribo-Zero-Gold kit (Illumina). Sequencing was performed using NextSeq Series High Output kit 2 × 75 bp (Illumina). Reads were quality controlled using FastQC, then mapped to (GRCh38/hg25) assembly of human genome and reads per gene quantified using STAR 2.5.2b (Dobin et al, 2013). Differentially expressed genes were identified using edgeR package (Robinson et al, 2010). Genes with at least 1 count per million in all samples were analysed for differential expression. Pairwise comparisons were made between input RNA and between polysomal RNA for samples transfected with pcDNA5, pcDNA 5 HA-CMTR1 WT or 2L/A. Plots comparing differential input or polysome mRNA abundance were drawn using the ggplot2 R package.

### RT–PCR

RNA was extracted using Trizol Reagent (Invitrogen). cDNA was synthesised using iScript cDNA Synthesis Kit (Bio-Rad). RT–PCR was performed using Quanta Biosciences SYBR Green. Primers: PRKDC_Fwd GAGAAGGCGGCTTACCTGAG, PRKDC_Rvr CGAAGGCCCGC-TTTAAGAGA, IGF2R_Fwd AGCGAGAGCCAAGTGAACTC, IGF2R_Rvr TCG-CTGTAAGCAGCTGTGAA, CAD_Fwd AGGTTTGCCAGCTGAGGAG, CAD_Rvr TAATGAGTGCAGCAGGGGTG, ATR_Fwd GGAGGAGTTTTGGCCTCCAC, A-TR_Rvr TGTGGCACTGCCCAGCTC, GCN1_Fwd CTTGTGCCCAAGCTGACAAC, GCN1_Rvr GCCCTGTGTCATCCTCTACG, MTOR_Fwd GAAGCCGCGCGAACCT, MTOR_Rvr CTGGTTTCCTCATTCCGGCT, CMTR1_Fwd CATTGCCCCATTTCA-CATTTGC, CMTR1_Rvr TCTTAGGCCCTGTGCATCTG.

### RNAseq data

RNAseq data can be found in the GEO database record GSE113573.

## Supplementary Information

## Acknowledgements

We thank the Cowling laboratory for assistance and Janusz Bujnicki laboratory, Frances Fuller-Pace, and Sara Ten-Have for discussions. Research was funded by Medical Research Council Senior Fellowship (VH Cowling) MR/K024213/1, Lister Institute Prize Fellowship (VH Cowling), Wellcome Trust Technology Platform award 097945/B/11/Z, and Medical Research Council, UK, Next Generation Optical Microscopy award MR/K015869/1.

### Author Contributions

F Inesta-Vaquera: conceptualization, data curation, formal analysis, validation, investigation, methodology, and writing—original draft, review, and editing.
V Chaugule: investigation and methodology.
A Galloway: data curation and formal analysis.
L Chandler: investigation.
A Rojas-Fernandez: investigation.
S Weidlich: investigation.
M Peggie: investigation.
VH Cowling: conceptualization, resources, data curation, formal analysis, supervision, funding acquisition, investigation, methodology, writing—original draft, project administration, and writing—review, and editing.

### Conflict of Interest Statement

The authors declare that they have no conflict of interest.

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
