## [Reviewer comments · Life Science Alliance]

DHX15 regulates CMTR1-dependent gene expression and cell proliferation

Francisco Inesta-Vaquera, Viduth K. Chaugule, Alison Galloway, Laurel Chandler, Alejandro Rojas-Fernandez, Simone Weidlich, Mark Peggie and Victoria H Cowling
DOI: 10.26508/lsa.201800092

Review timeline:	Submission date:	16 May 2018
	Editorial Decision:	31 May 2018
	Revision received:	4 June 2018
	Accepted:	5 June 2018

Report:

(Note: Letters and reports are not edited. The original formatting of letters and referee reports may not be reflected in this compilation.)

Please note that the manuscript was previously reviewed at another journal and the reports were taken into account in the decision-making process at Life Science Alliance.

1st Editorial Decision

31 May 2018

Thank you for submitting your revised manuscript entitled "DHX15 regulates CMTR1-dependent gene expression and cell proliferation". Your manuscript was previously reviewed at another journal, and the previous reviewer reports were confidentially transferred to us by the journal editors.

Publication of your work was supported by one reviewer, while another reviewer raised various issues. We had discussed with you prior to submission to our journal that for publication in Life Science Alliance a revised version addressing specific points of this reviewer needs to be provided. As mentioned to you before, we decided to seek arbitrating advice on the revised version of your manuscript, and we have now heard back from the advisor. Based on this input, we would be happy to publish your work in Life Science Alliance pending minor revision as outlined (=>) below.

The advisor assessed both your manuscript as well as your response to the specific concerns raised by reviewer #1 during peer-review at the other journal. The advisor thinks your work convincingly shows that CMTR1 interacts with DHX15, and appreciates the important in vitro data showing the effects of DHX15 on CMTR1 function in cap modification. The advisor thinks that a negative control with a 6Xhis-tagged irrelevant protein, or, a 6Xhis-tagged mutant of DHX15 that doesn't interact with CMTR1 should be included in this assay.

=> Since you have a negative control using an active CMTR1 mutant that cannot bind to DHX15, it is not mandatory to address this issue experimentally, but please comment on it.

The advisor finds it furthermore difficult to understand how the effect on a few RNAs can be reconciled with the big effect of DHX15 overexpression on GpppG O-2 methylation (Figure 3E).

=> We think that you discuss this careful enough in your discussion

The advisor thinks that competition between CTD and DHX15 could be easily tested.

=> We don't expect inclusion of additional experimental data as you discuss the potential competition.

The advisor thinks that you addressed most comments of reviewer #1 and that a control to show that

the E261Q mutant used lacks helicase activity is not needed. However, the advisor thinks that the request for more information on how endogenous mRNAs were analyzed for cap +1 methylation upon DHX15 overexpression is not sufficiently revised.

=> We agree that it is not clear based on the figure legend and your material and methods section how this measurement was done, and we assume that you used the cell lysate for an in vitro cap +1 methylation assay. Please state clearly what was done.

The advisor thinks that the finding that CMTR1 upregulates the helicase activity of DHX15 is likely to be physiologically more important than the effect of DHX15 on CMTR1 (though for what is not clear).

=> Please make sure to discuss this possibility carefully

1st Revision – authors' response

4 June 2018

The advisor assessed both your manuscript as well as your response to the specific concerns raised by reviewer #1 during peer-review at the other journal. The advisor thinks your work convincingly shows that CMTR1 interacts with DHX15, and appreciates the important in vitro data showing the effects of DHX15 on CMTR1 function in cap modification. The advisor thinks that a negative control with a 6Xhis-tagged irrelevant protein, or, a 6Xhis-tagged mutant of DHX15 that doesn't interact with CMTR1 should be included in this assay.

=> Since you have a negative control using an active CMTR1 mutant that cannot bind to DHX15, it is not mandatory to address this issue experimentally, but please comment on it.

Response: The suggestion is good, but we already have a CMTR1 mutant which retains methyltransferase activity but does not bind to DHX15. This mutant CMTR1 is not inhibited by DHX15. Identifying mutants of DHX15 which retain function but lose interaction with CMTR1 may be complicated by the fact that the domains of DHX15 exhibits quite complex functional/physical interactions which are outlined in the discussion.

The advisor finds it furthermore difficult to understand how the effect on a few RNAs can be reconciled with the big effect of DHX15 overexpression on GpppG O-2 methylation (Figure 3E).

=> We think that you discuss this careful enough in your discussion

The advisor thinks that competition between CTD and DHX15 could be easily tested.

=> We don't expect inclusion of additional experimental data as you discuss the potential competition.

The advisor thinks that you addressed most comments of reviewer #1 and that a control to show that the E261Q mutant used lacks helicase activity is not needed. However, the advisor thinks that the request for more information on how endogenous mRNAs were analyzed for cap +1 methylation upon DHX15 overexpression is not sufficiently revised.

=> We agree that it is not clear based on the figure legend and your material and methods section how this measurement was done, and we assume that you used the cell lysate for an in vitro cap +1 methylation assay. Please state clearly what was done.

Response: The amended sections are indicated in red

The results section now becomes:

“Transfection of HeLa cells with a titration of pCDNA5 HA-CMTR1 resulted in dose dependent O-2 methyltransferase activity in cell extracts (Figure 3e). Transfection with pCDNA5 GFP-DHX15 inhibited methyltransferase activity in cell extracts.”

Figure legend becomes:

e) HeLa cells were transfected with 5µg pcDNA GFP or GFP-DHX15 and 0, 0.4 or 2µg pcDNA5 HA-CMTR1, as indicated. 1 day after transfection, cells were harvested and 5µg cell extract was used in O-2 methyltransferase assay. % GpppG O-2 meth and standard deviation of duplicates reported.

Added to the methods:

In Figure 3e, when cell extracts were used in the assay, cells were lysed in lysis buffer (as above) and 5µg cell extract was used in the methyltransferase reaction.

The advisor thinks that the finding that CMTR1 upregulates the helicase activity of DHX15 is likely to be physiologically more important than the effect of DHX15 on CMTR1 (though for what is not clear).

=> **Please make sure to discuss this possibility carefully**

Response: The penultimate paragraph of the discussion now becomes (amended section indicated in red):

“Ultimately we want to understand how the relationship between DHX15 and CMTR1 impacts on cell function. This is complex due to the multifunctional nature of both enzymes. The impact of the DHX15-CMTR1 interaction will depend on the relative expression of the enzymes and their other interacting partners, which also influence enzyme activity and localisation. The impact of the DHX15-CMTR1 interaction will also depend on underlying cell physiology, including relative dependency on O-2 methylation, splicing, rRNA processing and translation. Of note, CMTR1 is an interferon-regulated gene whereas DHX15 is not, and therefore interferon signalling alters the ratio of these factors. In this study we focus on the biological impact of DHX15 repression of CMTR1 methyltransferase activity and function. We stress that, in certain contexts, the impact of CMTR1 activation of DHX15 helicase activity may be equivalently or more biologically important, given the role of DHX15 in RNA processing.”

2nd Editorial Decision

5 June 2018

Thank you for submitting your Research Article entitled "DHX15 regulates CMTR1-dependent gene expression and cell proliferation". I appreciate the introduced changes, and it is a pleasure to let you know that your manuscript is now accepted for publication in Life Science Alliance. Congratulations on this interesting work.